# Natural Fillers as Potential Modifying Agents for Epoxy Composition: A Review

**DOI:** 10.3390/polym14020265

**Published:** 2022-01-10

**Authors:** Natalia Sienkiewicz, Midhun Dominic, Jyotishkumar Parameswaranpillai

**Affiliations:** 1Institute of Polymer and Dye Technology, Faculty of Chemistry, Lodz University of Technology, Stefanowskiego 16, 90-537 Lodz, Poland; 2Department of Chemistry, Sacred Heart College (Autonomous), Kochi 682013, Kerala, India; midhundominic@shcollege.ac.in; 3Department of Science, Faculty of Science & Technology, Alliance University, Chandapura-Anekal Main Road, Bengaluru 562106, Karnataka, India; jyotishkumarp@gmail.com or

**Keywords:** natural fillers, green composites, seed fibers, fruit fibers, grass fibers

## Abstract

Epoxy resins as important organic matrices, thanks to their chemical structure and the possibility of modification, have unique properties, which contribute to the fact that these materials have been used in many composite industries for many years. Epoxy resins are repeatedly used in exacting applications due to their exquisite mechanical properties, thermal stability, scratch resistance, and chemical resistance. Moreover, epoxy materials also have really strong resistance to solvents, chemical attacks, and climatic aging. The presented features confirm the fact that there is a constant interest of scientists in the modification of resins and understanding its mechanisms, as well as in the development of these materials to obtain systems with the required properties. Most of the recent studies in the literature are focused on green fillers such as post-agricultural waste powder (cashew nuts powder, coconut shell powder, rice husks, date seed), grass fiber (bamboo fibers), bast/leaf fiber (hemp fibers, banana bark fibers, pineapple leaf), and other natural fibers (waste tea fibers, palm ash) as reinforcement for epoxy resins rather than traditional non-biodegradable fillers due to their sustainability, low cost, wide availability, and the use of waste, which is environmentally friendly. Furthermore, the advantages of natural fillers over traditional fillers are acceptable specific strength and modulus, lightweight, and good biodegradability, which is very desirable nowadays. Therefore, the development and progress of “green products” based on epoxy resin and natural fillers as reinforcements have been increasing. Many uses of natural plant-derived fillers include many plant wastes, such as banana bark, coconut shell, and waste peanut shell, can be found in the literature. Partially biodegradable polymers obtained by using natural fillers and epoxy polymers can successfully reduce the undesirable epoxy and synthetic fiber waste. Additionally, partially biopolymers based on epoxy resins, which will be presented in the paper, are more useful than commercial polymers due to the low cost and improved good thermomechanical properties.

## 1. Introduction

Epoxy resins are one of the versatile thermosetting polymers widely employed for different applications such as construction, coating, automobile, aerospace, and structural adhesives. Essentially, epoxy resins are low molecular weight liquids with two or more epoxide functional groups. These epoxide groups in epoxy resins can easily react with a wide range of curing agents/hardeners such as acids, anhydrides, alcohols, and amines. During curing, the low molecular weight liquid epoxy resin increases in length, makes branches, and finally becomes a high molecular weight solid material with a three-dimensional network structure [1,2]. The completely cured epoxy resins are known for their high strength, modulus, thermal stability, low shrinkage, chemical resistance, and dimensional stability [3]. However, the cured epoxy samples are highly brittle and are susceptible to moisture absorption. The high brittleness of the epoxy system was due to their high crosslink density. Xian et al. [4] reported that the crosslinked epoxy resins may undergo hydrolytic degradation or network hydrolysis when exposed to a humid environment. This may cause the following changes in the epoxy network such as plasticization, reduction in glass transition temperature (*T_g_*), and the degradation in mechanical properties when exposed to wet aging conditions.

The current demand for high-performance green composites has resulted in the use of natural fillers as a reinforcement in the epoxy matrix. The USA, European, and other government agencies encourage the use of green composites with a high number of natural resources [5,6,7,8,9]. Studies have shown that the incorporation of various natural fillers in epoxy resin improved its thermomechanical properties. Many automobile companies are currently using green fibers in their automotive products, since there is a global emergency of low CO_2_ emission [6,7,8,9].

The need of utilizing green materials for a sustainable environment has caused an upsurge in the use of natural materials in the composite industry because the sustainability of the products during processing and end of life is important. Natural fillers are sustainable potential materials as reinforcing agents for different polymer matrices in varying applications such as automobile, construction, aerospace, toys, defense, sporting goods, and electronic applications [10]. The key objectives of natural filler in the composite industry were to reduce the cost as well as improve the processing, properties, and environmental friendliness. The cost of different natural fibers and E-glass fiber per kilogram is given in Figure 1 [11]. From the figure, when compared with E-glass fiber, all the natural fibers are cheaper.

The other advantages of natural fibers are their easy availability, easy manufacturing process, less energy consumption, renewability, and good mechanical properties. These advantages make natural fillers an alternative to traditional fillers in many applications such as construction and infrastructure, furniture, and rotor blade materials [10]. Based on their origin, natural fibers are plant-based, animal-based, and bacterial-based [12,13,14]. However, plant-based natural fibers are preferred due to their abundance, easy availability, and low cost. The plant fibers can be obtained from bast (flax, hemp, and ramie), leaves (pineapple, abaca, and sisal), fruits/seeds (cotton and coir), stalk (wheat, rice, and oats), and grass (bamboo and bagasse) [15,16]. Plants fibers from different sources have varying amounts of cellulose, hemicellulose, and lignin content; hence, it is worth characterizing plant fibers from different sources and using them as a reinforcement in the polymer matrix [17]. Note that a high content of cellulose in fibers is recommended for the fabrication of composites with high strength and modulus. Table 1 shows the fiber source, world production, cellulose content, tensile strength (TS), Young’s modulus, and density [17]. The cellulose content of ramie, flax, and jute was the highest. Among the various natural fibers, flax showed the highest tensile strength, modulus, and lowest equilibrium moisture content [17].

Plant-based fillers such as banana fiber, hemp, sisal, pineapple, bamboo, flax, peanut particles, etc., have been widely used as reinforcement for various polymer matrices. Studies have shown that plant fibers are an excellent replacement for carbon and glass fibers in many semi-structural applications [18]. Recently, the isolation of nanocellulose from plant fibers has received huge attention due to the ease of preparation, good strength, modulus, and crystallinity. The nanocellulose can be used for the difference of application as a replacement for carbon-based fillers, metallic fillers, and polymeric nanofillers [19]. It is important to point out that the natural fillers/fibers in macro, micro, and nanoscales have a significant impact on the mechanical properties of the polymer matrix [20,21,22]. However, a few shortcomings have limited its widespread application in the epoxy composite industry. These are poor water sensitivity, UV radiation, poor bonding, flammability, dissimilar chemical nature, and lower mechanical properties compared with synthetic fibers. The poor moisture resistance and poor bonding or interfacial properties of the natural fillers are due to the hydrophilic nature because of the presence of hydroxyl groups present in them. Moreover, the moisture or water may enter inside more easily through the polymer/fiber interface and affect the short- and long-term properties of the epoxy composites. Thus, there is a high chance of the degradation of epoxy composites when used for outdoor applications [23]. These shortcomings can be overcome by the physical treatment of natural filler (corona treatment, plasma treatment) [24,25], chemical treatment of natural filler (silane, alkali, benzoylation, etc.) [26,27,28,29,30], and modification of natural fillers with coupling agents [31].

The utilization of physical and chemical treatment has shown reduced hydrophilicity and improved thermomechanical performance of the fibers. However, chemical treatment is not environmentally friendly and is also not cost-effective; therefore, green, ecofriendly, naturally-derived materials such as tannic acid are recently employed for the surface functionalization of natural fibers [32]. The physical and chemical treatment of the fibers, and the addition of coupling agents, have been reported to have an effect not only on the surface morphology but also on the hydrophilicity, thermomechanical, and water absorption properties of the composites. The studies reported enhanced short- and long-term properties by improving the mechanical, fatigue, creep, thermal, and water resistance properties of the natural fibers-reinforced epoxy composites after various physical and chemical treatments [33,34,35,36,37,38]. The studies also reported that the factors such as filler content, filler size, and filler geometry also have a significant influence on the properties of the composites [37,38]. 

Improved composite materials can be fabricated with the addition of natural fibers in the epoxy matrix. Thus, the use of natural materials is essential for the development of low-cost semi-structural epoxy composites and also to reduce undesirable epoxy, synthetic fiber waste, and CO_2_ emissions. Therefore, in this study, we have critically reviewed the possibilities of waste natural materials in the epoxy composite industry. A schematic of various natural fibers used for the development of natural fiber/filler-reinforced epoxy composite is given in Figure 2.

## 2. Post-Agricultural Waste Powder Material Filled Epoxy Composites (Seed and Fruits)

Post-agricultural waste materials are of interest for the manufacturing of low-cost composites due to their low price, easy availability, and sustainability. Natural materials such as walnut waste shell, tamarind shell, peanut shell powder [38,39,40], etc. have been acquired wide interest for their application in composite technology. Baig and Mushtaq [39] studied the influence of tamarind shell powder on the mechanical properties of epoxy composites. The photograph of tamarind shell and tamarind shell powder is shown in Figure 3. The 50/50 and 30/70 epoxy/tamarind shell powder compositions were prepared. The composite with a 30/70 composition showed the highest TS, tensile modulus (TM), and hardness, while the 50/50 composition showed the highest flexural strength (FS) and flexural modulus (FM). The water absorption is more for 30/70 compositions. In an interesting study, Prabhakar et al. [40] used NaOH treated waste peanut shell powder as a reinforcement for DGEBA epoxy resin. The researchers observed that the incorporation of waste peanut shell powder improved the TS, TM, and thermal stability.

Salasinska et al. [41] used walnut shell waste powder as a modifier for epoxy composites. The walnut shell waste filler was milled, and a particle size in the range of 32–120 µm was prepared. The concentration of the filler used in the epoxy matrix was 20%, 30%, 40%, and 50%. The FTIR spectrum revealed that the curing process was modified in the presence of fillers. The study reported a reduction in moisture content in epoxy composite with the incorporation of walnut shell powder; however, some moisture will always be present depending on the environmental conditions. The mechanical properties such as TS and impact properties were reduced with the incorporation of walnut shell powder due to the poor dispersion of particles in the matrix. On the other hand, YM, hardness, and storage modulus (SM) were increased; the authors claim that this is due to the catalytic effect of OH groups present in fibers on the epoxy reaction kinetics, which may result in a higher crosslink density of the samples. The thermal stability of the composites (5% mass loss, 10% mass loss, and residual mass) was also increased due to the higher thermal stability of the walnut shell waste powder. In a more recent study, Albaker et al. [42] used treated walnut shell waste powder as a filling material for epoxy matrix. The walnut shell was milled, sieved, and then alkali-treated. Then, the alkali-treated filler was treated with three different organic acids such as citric acid, oxalic acid, and formic acid. The treatment increases the percentage of cellulose, while it reduces the percentage of hemicellulose and lignin. The concentration of the treated filler used in epoxy was 10%, 20%, 30%, 40%, and 50%. The composites prepared showed an increased TS and YM with comparable hardness, while the elongation at break was reduced. The optimum concentration of the treater filler was 20 wt%, irrespective of the type of treated fiber. On the other hand, the thermal stability of the composite was slightly reduced, while the water absorption was marginally increased. Barczewski et al. [43] studied and compared the feasibility of sunflower husk, hazelnut shell, and walnut shell as waste agricultural filler for developing low-cost epoxy composites. Before incorporating in the epoxy matrix, the sunflower husk, hazelnut shell, and walnut shell were ground, as shown in Figure 4. The content of the filler used in the composite was 15%, 25%, and 35%. The TM, FM, and hardness of the composites were increased, while the TS, FS, and IS were reduced, with the increasing addition of ground waste fillers. Among the composites, hazelnut shell-based organic waste filler showed the best properties. Thus, the hazelnut shell filled composite was most favorable for developing eco-friendly composites 

Sathishkumar et al. [44] studied the dry sliding wear and friction performance of epoxy composites containing cashew nutshell powder. The cashew nutshell powder was prepared from cashew nut seed, and the powder was treated with 5% NaOH solution to remove the weak amorphous phase from it. The concentration of the NaOH treated filler used in epoxy resin was 5%, 10%, 15%, 20%, and 30%. It is also worth noting that the coefficient of friction and specific wear rate was low for the epoxy sample containing 30% treated filler as a result of better bonding and uniform distribution of the filler. Since the nutshell is a waste part cashew and is fully biodegradable, it is worth using it for the development of epoxy composites for tribological applications. Shakuntala et al. [45] fabricated wood apple shell particulates composites with improved mechanical and wear resistance. The wood apple shell particulates of ca. 212 μm from wood apple shells were prepared by crushing and ball milling. The density and void content of the epoxy matrix was decreased with the incorporation of wood apple shell particulates. The TS, FS, interlaminar shear strength, and SM was increased with the incorporation of wood apple shell particulates in the epoxy network, and the best results were observed for 15% filler content. The improved mechanical properties are due to the good wetting of the filler by the epoxy resin. However, a drop in mechanical properties was observed for 20 wt% filler contents, because at higher loading, it is more difficult for effective wetting of the filler by the epoxy polymer, leading to poor interfacial interaction between the filler and polymer. The erosion wear properties of the composites were tested at different impingement angles. The composites showed lower erosion wear properties compared to neat epoxy, and the 10% and 15% composites showed the lowest erosion wear properties. Irrespective of the samples prepared, the erosion rate was maximum at an impingement angle between 45° and 60°. The photograph of wood apple fruit and wood apple shell, SEM images of the wood apple shell particles, and erosion wear behavior of the composites at different impingement angles are shown in Figure 5.

Soumyalata et al. [46] studied and compared the effect of coconut shell powder and rice husk powder as a reinforcing filler in epoxy composites. The mechanical properties, specific gravity, and water absorption properties of the composites were studied. Compared to rice husk powder, the coconut shell powder-reinforced epoxy composite gives higher TS, FS, impact strength (IS), and hardness (shore D). The water absorption resistance was also reported to be higher for coconut shell powder-reinforced epoxy composite. The reported specific gravity for both the composites is similar: 1.21 and 1.2, respectively. The comparative study showed that the coconut shell powder-reinforced epoxy composite was superior to the rice husk-reinforced epoxy composite. Similarly, Salleh et al. [47] reported an increased Izod impact strength of epoxy composites with the incorporation of Komeng coconut carbon fiber. Thus, many studies reported encouraging results for the possible use of post-agricultural waste powder in the composite industry. However, the impact of degradation of fruit and seed fillers in epoxy composites during their long-term service time has not been considered. The lack of information on the degradation of the composites limits the use of fruit and seed fillers in many service applications. Therefore, more experimental, theoretical, and prediction studies on the fruit and seed fillers-reinforced epoxy composites are required to ensure better serviceability [48].

## 3. Grass Fiber-Based Epoxy Composites

Napier, bamboo, and bagasse are widely used in construction applications. Kommula et al. [49] studied the effect of incorporation of untreated and alkali-treated Napier grass in the epoxy matrix. Randomly oriented short and long unidirectional fibers were used for the fabrication of the composites. The fiber loading used was 10%, 20%, and 30%, and the NaOH concentration used was 5%, 10%, and 15%. The TS, TM, FS, FM, and IS of the composites were improved with fiber addition and NaOH treatment. Irrespective of the type of fiber (randomly oriented short and long unidirectional fibers), the best properties were observed for 20% filler content with 10% alkali treatment. The improvement in properties was due to the improved interfacial adhesion between the fiber and polymer after the alkali treatment. Note that long fibers show superior properties compared to short fibers. The water absorption and chemical resistance of the composite were also reduced with alkali treatment. Based on the properties, the researchers recommend the alkali-treated Napier grass treated epoxy composites for semi-structural applications. 

Agricultural waste bagasse was treated with triglycidyl isocyanurate (TGIC) and 9,10-dihydro-9-oxa-10-phosphaphenanthrene-10-oxide (DOPO) to graft nitrogen and phosphorous-containing compounds onto its surface [50]. Then, the modified bagasse was incorporated with the epoxy network to fabricate fire-retardant epoxy composites. The reaction process of epoxy/modified bagasse is shown in Figure 6. The study revealed that the modified bagasse improved the initial pyrolysis temperature, suppressed smoke, and showed excellent flame retardancy in both UL94 and LOI tests. The mechanism for the increased flame retardancy was biochar formation; this is because DOPO produces phosphoric acid during thermal heating and promotes biochar formation. Thus, halogen-free flame retardant could be developed by using bio-based flame retardant. 

Fiore et al. [51] studied the feasibility of fibers from Arundo Donax, a noon wood plant with very fast growth as a reinforcement for epoxy composites. To make fine fillers, the Arundo Donax plants were collected, dried, and ground by a grinding machine and later sieved by a sieving machine. The fibers were not chemically treated to keep the cost low. The following inference was observed from the study. With the incorporation of fillers, the void content increases, TM increases, and FM marginally changes, while the TS and FS were reduced. The drop in TS and FS was due to (i) the hydrophilic nature of the fillers and (ii) high void content. The dynamic mechanical analysis shows no variations before *T_g_*, but after *T_g_*, the SM being higher may be due to the presence of rigid filler in the rubbery matrix. Based on the results, the researchers suggested the use of Arundo Donax filler-based epoxy composites for semi-structural applications. Kumar et al. [52] treated bamboo filler with NaOH, and the treated filler was incorporated in the epoxy network to improve the mechanical properties of epoxy composites. The filler content used was 2.5%, 5%, 7.5%, 10%, and 12.5%. The void content in the composites was increased with the increase in the filler content for both untreated and treated filler. However, the treated composites reported minimum void content. The TS and FS were increased with the increase in the filler content (up to 10%) for both untreated and treated filler. Another very interesting example for green epoxy composites was an efficient and eco-friendly solution proposed by Singh et al. [32]. Here, the naturally derived tannic acid is used to modify bamboo micron fibers. The modified fibers are used as a reinforcement to prepare high-performance epoxy composites. The results showed that the incorporation of 5 wt% treated fibers in the epoxy matrix enhanced the stress intensity factor and critical strain energy release rate by ≈60% and ≈212%, respectively.

## 4. Bast and Leaf Fibers Modified Epoxy Composite

Maleque et al. [53] used pseudo-stem woven banana fabric to enhance the mechanical properties of epoxy composites. The TS, FS, TM, FM, and IS of the epoxy composites increased considerably. The SEM micrographs of the composites revealed good interfacial bonding between the fiber and epoxy matrix. Masiewicz et al. [54] used three different natural materials such as collagen, hemp fibers, and pepper powder as a modifier for epoxy matrix and studied the changes in gel time and mechanical properties. For the preparation of epoxy composites, hemp fibers and pepper powder were mixed directly with epoxy resin, while collagen was first dissolved in ethylene glycol followed by mixing in epoxy resin. The curing was done for 24 h at room temperature and post-curing at 80 °C for 3 h. The filler content used was 5–20%. The study reported an increase in gel time with the incorporation of fillers. Irrespective of the type of filler, the IS, flexural strain, and critical stress intensity were increased, and maximum values were reported for composites containing 5% of filler content. Among the composites studied, the IS and critical stress intensity were maximum for 5% pepper-modified epoxy composites, and increases of 270% and 330% were observed respectively compared with neat epoxy. The impact strength of epoxy composites containing 5–20% natural filler is shown in Figure 7. The beneficial effects in improving the properties of epoxy composites point out the possible use of green fillers as an alternative to synthetic fibers for composite applications. 

Ridzuan et al. [55] presented the effects of pineapple leaf (PALF), Napier, and hemp fibers on the scratch resistance of epoxy composites. The effect of the fillers on the horizontal load, coefficient of friction, penetration depth, scratch hardness, and scratch observation were studied. The concentration of filler used was 5, 7.5, and 10 wt%. The scratch resistance and coefficient of friction were highest for the higher wt% of bio-fillers, and among the composites, the Napier-filled epoxy composites showed better scratch resistance and coefficient of friction compared to the pineapple leaf and hemp fiber-filled epoxy composites. In addition, the penetration depth is lower for Napier than the pineapple leaf and hemp bio-filled epoxy composite. These results support the highest scratch resistance for Napier-modified epoxy composites. Later, Ridzuan et al. [56] fabricated jute/epoxy, kenaf/epoxy, and Napier-filled epoxy composites and studied the water absorption and dielectric properties of the composites. The concentration of the fiber used was 7%, 14%, and 21%. The time of immersion and the change in the fiber content has a significant effect on the water absorption properties of the composites. For all three composite series, the water absorption increases with an increase in the concentration of fiber content and with an increase in the time of the experiment. A saturation in water absorption in all these composites was observed at 500 h. The kenaf and Napier-filled composites showed lower water absorption than jute fiber-filled composites. The dielectric constant of the composites in dry and wet conditions was studied. The Napier-reinforced epoxy matrix showed the lowest dielectric constant between 12.4 and 16.66 GHz.

Shah et al. [57] prepared epoxy composites containing 0.5, 1.0, 1.5, and 2.0 wt% Acacia Catechu powder. Fourier transform infrared spectroscopy confirmed increased cure conversion in filled epoxy. Adding a very small amount of filler (1.0 wt%) resulted in a 14% increase in the FS and 94% improved IS due to the modification in morphology and crosslink density. Additionally, the aromatic tannin phenol structures of Acacia Catechu improved the thermal stabilities of epoxy composites. In addition, the scanning electron microscope analysis showed shear banding phenomena between filler particles and epoxy resin, which caused increased toughness. Gargol et al. [58] fabricated epoxy composites containing varying amounts of waste hemp fibers. The curing agent used was triethylenetetramine and the fiber content used was 0, 5, 10, 15, and 20 wt%. The samples were cured at room temperature for 10 h. The intermolecular interaction between the hemp fiber, epoxy, and the curing agent was schematically represented in Figure 8. The FTIR studies of the composites reported the absence of the peak at 904 cm^−1^ due to the absence of epoxide ring because of the complete curing of the composite samples. Thermogravimetric analysis showed improved thermal stability for the composite samples compared to neat polymer and fiber. The DSC study reported a major exothermic peak at slightly above 300 °C due to the degradation of the composite samples and was marginally improved for the composite samples compared to the pure matrix. On the other hand, the tensile, flexural, and hardness values of the epoxy composites were reduced with the incorporation of fibers. Maleki et al. [59] studied the effect of drilling on flax epoxy composites using three different drill bits: a twist drill, CoroDrill 854, and CoroDrill 856. The thrust force was increased with an increase in spindle speed and feed. Among the three different drill bits, the twist drill has the lowest thrust force at different conditions. Furthermore, the delamination was lowest for the twist drill, and the optimum spindle speed and feed were 1500 rpm and 0.2 mm/rev, respectively.

## 5. Other Natural Fillers Reinforced Epoxy Composites

A new study on the utilization of nano-oil palm ash as filler material for epoxy composite production was presented by Khalil et al. [60]. The study focused on the effect of the filler content on the physical, mechanical, and thermal properties of epoxy composites. The density of the nano-structured oil palm ash-filled epoxy samples was increased with increasing filler loading. On the other hand, the void content showed a low value for the 1% composites; this was followed by a moderate increase, only 5% composite showed higher void content than the pure epoxy matrix. The mechanical properties such as TS, TM, FS, FM, and thermal stability were increased with filler loading, and the best results were achieved at 3% filler. In addition to using fillers from plant sources, in recent years, researchers also focused on the use of waste fibers such as CF, eggshell, seashell, etc., as fillers for the epoxy composites [61,62]. Bessa et al. [63] studied the effect of the addition of CF on the thermal resistance and noise reduction of epoxy composites. Three different compositions—20/80, 30/70, and 40/60 (epoxy/CF)—were prepared. The researchers observed that the thermal resistance and acoustic insulation properties were maximum at the highest CF content, i.e., 20/80 epoxy/CF composite. This was due to the hollow structure of the CF, i.e., the internal hollow channel in the CF contributes to thermal and acoustic reduction. Vijayan et al. [64] developed CF-incorporated epoxy coating with improved performance. The pure epoxy and CF incorporated epoxy were coated on carbon steel and subjected to an accelerated salt immersion test. The progress of corrosion was evaluated. During the initial days of the test, no corrosion was observed for both epoxy/CF and epoxy coatings. However, after two weeks, corrosions are observed in the coating, but the corrosion is least affected in epoxy/CF coatings. This is because in neat epoxy coatings, many holes and cavities are generated during coatings, which results in localized corrosion called pitting corrosion. However, pitting corrosion is less in number in the case of epoxy/CF composites due to the absence of holes and cavities because of the better interfacial interaction between the CF and epoxy matrix. Thus, the CF acts as an effective anti-corrosion agent.

Abdelmalik et al. [65] studied the variations in TS and insulation properties of the epoxy composites with the addition of eggshell powder. The eggshell was washed with water, acetone, and methanol and dried. The cleaned eggshell was ball milled to have eggshell powder with ca. 75 µm. The tensile strength of epoxy composites was increased with the increasing addition of eggshell powder, and the best results were observed for 4 wt% filler, which was followed by a reduction for the composite with 5% filler content due to the agglomeration of the particles. The polymeric insulation properties were reduced with the incorporation of eggshell powder, and the minimum value was observed for 3% filler; a marginally higher electrical conductance was observed for 4 and 5% filler content. It is worth pointing out that all the composites fabricated reported lower electrical conductance compared to neat epoxy. Fombuena et al. [66] fabricated and characterized bioepoxy composites filled with seashell wastes. The researchers collected seashell from the coast of Valencia (Spain). It is washed with water and 4% NaOH to remove the impurities and milled and sieved to powder of 250 µm. The XRD profile reported both aragonite and calcite crystal phases in seashell (CaCO_3_) powder. The TGA thermogram of the seashell powder reported two-step degradation; a minor degradation at ca. 250 °C is due to the degradation of organic content, and the main degradation at approximately 817 °C is due to CaCO_3_. It was observed that the incorporation of seashell powder improved the FM, hardness, and *T_g_* of the composites. 

## 6. Hybrid Composites

Hybridizing various natural fillers with synthetic fibers is one of the recent approaches adopted by researchers to get the best thermomechanical performance of composites. The natural fibers lack strength, modulus, and durability compared with synthetic fibers such as glass, aramid fiber, Kevlar, and carbon. In addition, the other drawbacks of natural fibers such as moisture absorption, flammability, poor bonding with polymer matrix, etc., can be overcome by hybridizing with synthetic fibers. Note that synthetic fibers have good mechanical properties, long-term durability, negligible water absorption, excellent compatibility, and good thermal resistance. As a result of these superior properties, they are widely used in the epoxy composite industry for favorable mechanical properties, fatigue properties, and durability [67,68,69,70,71,72]. In this method, the properties of two or more fillers can be combined, and synergy in the performance can be achieved. The advantages of hybridizing natural fiber with synthetic fibers are cost reduction, flexibility in tailoring composite design, and enhancement in thermomechanical properties [67,73]. Xian et al. [73] fabricated epoxy/carbon (C)/flax (F) hybrid composites with different stacking sequences and studied the effect of stacking sequences (CFFFC, FCFCF) on the energy absorption and damping properties. The composite with a sandwich structure (CFFFC) showed higher energy absorption than FCFCF, which was due to the high strength of the carbon outer layer and better bonding between the flax fabric inner layers. On the other hand, the damping coefficient is greatest for alternately stacked (FCFCF) composites, because the deformation of flax is higher than carbon. In a recent study, Xian et al. [74] developed two types of (fiber random hybrid (RH) and core–shell hybrid (CH)) epoxy carbon/glass fiber hybrid composite rods. The composites were subjected to a combined water immersion and bending test for 360 days. The random hybrid (RH) composite rod showed greater strength after the aging studies. The long-term prediction in bridge service environments was conducted and revealed better serviceability for the RH composite rod.

Prabhu et al. [75] hybridized sisal and waste tea leaf fibers with glass fibers that were incorporated in the epoxy matrix. The sisal and waste tea leaf fibers were selected because of their sound absorption properties. Before applying the fibers, they were treated with 5% of NaOH to remove non-cellulosic components from the fiber. The weight of the glass fiber in all composites was kept at 10%, while the composition of sisal and waste tea leaf fibers was varied, but their total weight is fixed at 30%. Among the hybrid composites, the composite with 20% sisal, 10% waste tea leaf fiber, and 10% glass showed the best TS and FS. The impact energy values are also high for the composite with 20% sisal, 10% waste tea leaf fiber, and 10% glass. The alkali treatment and hybridization increase the interfacial adhesion between the fiber and epoxy matrix phase, which in turn increases the mechanical properties. The sound absorption coefficient of the composites was tested in the frequency range 63 to 6300 Hz, and the maximum sound absorption coefficient was observed for the hybrid composites. The increased mechanical properties and sound absorption coefficient suggest the potential application of the epoxy hybrid composites containing glass/sisal and tea powder in automobiles components, soundproofing materials, and interior paneling 

Wang et al. [76] studied the effect of hybridizing flax fiber sheets with glass fiber sheets in epoxy composites. Different combinations of hybrid composites with eight layers were fabricated, and among the composites prepared, G2F4G2 (two glass fiber sheets each sandwiched over four layers of flax) showed very high FS and FM compared to neat flax fiber-reinforced epoxy composites. The G2F4G2 composite also reported an 84% increase in SM with an 8 °C increase in *T_g_* compared to neat flax fiber-reinforced epoxy composites. Fatinah et al. [77] studied the effect of the addition of untreated Napier fiber, 5% NaOH-treated fiber, and hybridized untreated Napier fiber/glass fiber in the epoxy network. The amount of filler was kept constant at 25%. Researchers observed the reported TS and TM in the order untreated Napier fiber/glass fiber > 5% NaOH treated fiber > untreated fiber. Mansor et al. [78] used an analytical hierarchy process and found that kenaf fiber is the best among the natural fibers for the fabrication of hybrid composites with glass for the designing of a brake lever in automotive. Elkhouly et al. [79] studied the effect of date seed filler on the abrasive wear of glass fiber–epoxy composites. The study showed a very good enhancement in wear resistance and toughness of glass fiber/epoxy composites with the addition of date seed filler. The date seed filler is effective at reducing the cost of the composites but also has good wear resistance and toughness.

Zhan and Wool [80] fabricated hybridized waste CF and glass fiber epoxy composites with lower density. The CF-reinforced epoxy composites possess lower strength and modulus, while replacing a part of the CF with glass fiber improved the modulus and strength. Thus, hybridization is a useful way to make use of waste CF in composite applications. Nguyen et al. [81] fabricated a prototype of a bike silencer using hybrid epoxy composites containing glass and CF. Kocaman and Ahmetli [82] investigated the influence of acrylated soybean oil (AESO) along with banana bark and seashell in the mechanical properties of epoxy composites. The concentration of acrylated soybean oil used is 50%. Note that acrylated soybean oil was used as a co-matrix. The seashell-containing epoxy systems exhibit higher TS, e-modulus, and hardness than banana bark epoxy composites. Later, Ozkur et al. [83] studied the effect of blending soybean oil in epoxy resin and reinforced with jute woven fabric. The composites showed maximum impact strength over 50% soybean oil. While the maximum TS was reported at 30% soybean oil, the maximum FS was reported at 20% soybean oil. Both TS and FS decrease above 30% soybean oil. Thus, soybean oil has a significant influence on the performance of composites.

## 7. Current Challenges and Limitations

The performance of the green composites in their service life depends on various environmental conditions such as rain, moisture, temperature, and UV light. In the case of fiber-reinforced epoxy composites, the moisture or water may enter inside more easily through the polymer/fiber interface; therefore, degradation in mechanical properties is inevitable during the service life. The conditions such as increase in temperature and UV irradiation accelerate the chain scission of the polymers and filler degradation. Therefore, the degradation, mechanism, and understanding of the long-term performance of the composites are vital for the successful implementation of the composite materials in various applications. However, the lack of information on long-term performance limits the use of natural fiber in many service applications [84,85]. This highlights the usefulness of new technologies such as accelerated weathering and wet aging studies of epoxy composites to evaluate the long-term durability of the composites in various environmental conditions. The modeling studies of water transport, thermal degradation, photodegradation, etc., would be worth understanding the long-term performance of the composites to get a conclusive overview of the long-term performance of the composites.

## 8. Conclusions

This review presented that the addition of bio-fillers of natural origin positively changes and improves the performance of the epoxy composites. These composite materials are becoming more attractive due to their suitability, ecofriendly nature, availability of raw materials, low cost, and good performance. A large range of new materials can be designed using natural filler-modified epoxy composites for versatile applications. 

In this paper, cashew nuts powder, hemp fibers, bamboo fibers, palm ash, waste tea fibers, banana bark fibers, coconut shell powder, rice husks, date seed, pineapple leaf, and other bio-fillers as an enhancement for epoxy resin as matrix are reviewed. The literature reports a favorable change in the properties of the neat epoxy system with the incorporation of natural fibers. It is proved that cost-effective composites can be developed by using various natural fillers. The recent research results caused an upsurge in the fabrication of lightweight hybrid composites with more environmentally friendly materials and have been employed in automobile and construction parts. However, the poor water sensitivity, poor bonding, flammability, and lower mechanical properties compared with synthetic fibers limited its widespread application in the epoxy composite industry. The chemical and physical treatments of natural fibers are usually adapted to overcome the drawbacks of natural fiber. However, the lack of information on long-term performance limits the use of natural fiber in many service applications. Therefore, more experimental, theoretical, and modeling studies are much needed to predict the serviceability. 

## Figures and Tables

**Figure 1 polymers-14-00265-f001:**
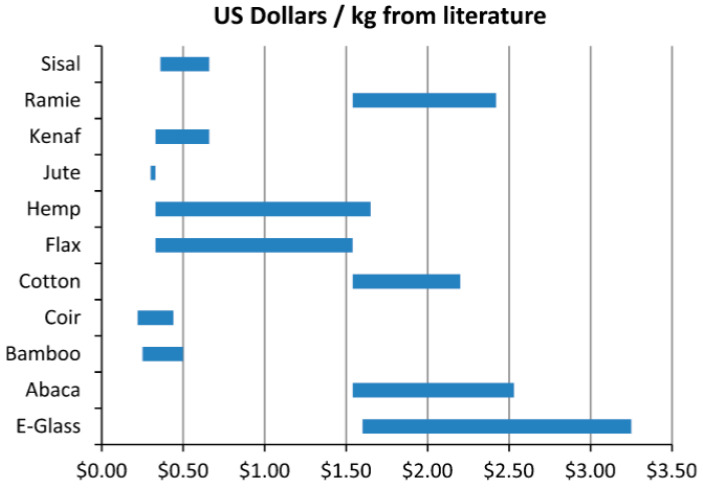
The cost of different natural fibers and E-glass fiber per kilogram in US dollars [11] (reproduced with thanks from Elsevier, License Number: 5206321182202).

**Figure 2 polymers-14-00265-f002:**
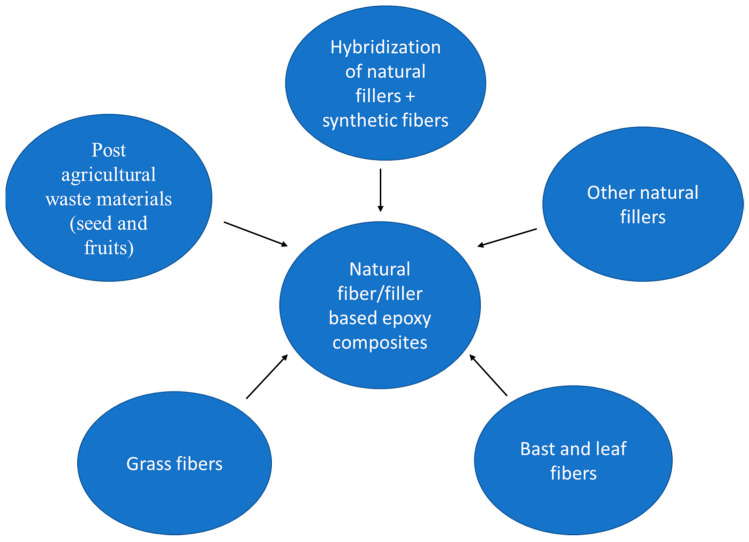
A schematic of various natural fibers used for the development of natural fiber/filler-reinforced epoxy composite.

**Figure 3 polymers-14-00265-f003:**
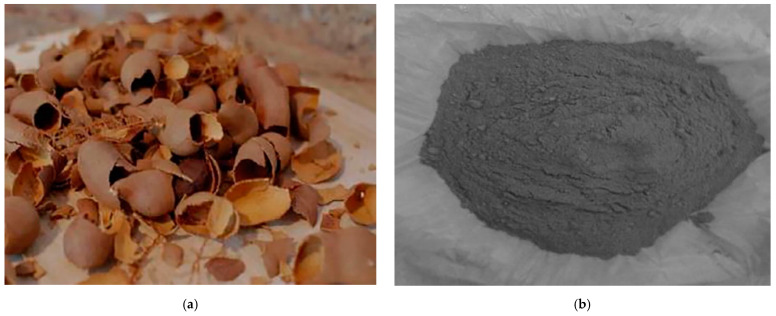
The photograph of (**a**) tamarind shell and (**b**) tamarind shell powder [39] (reproduced with thanks from Elsevier, License Number: 5206490581705).

**Figure 4 polymers-14-00265-f004:**
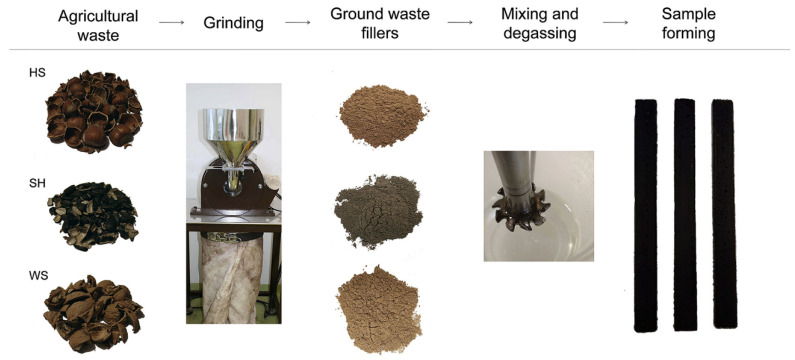
Fabrication steps of sunflower husk, hazelnut shell, and walnut shell-based epoxy composites [43] (reproduced with thanks from Elsevier, License Number: 5206741042858).

**Figure 5 polymers-14-00265-f005:**
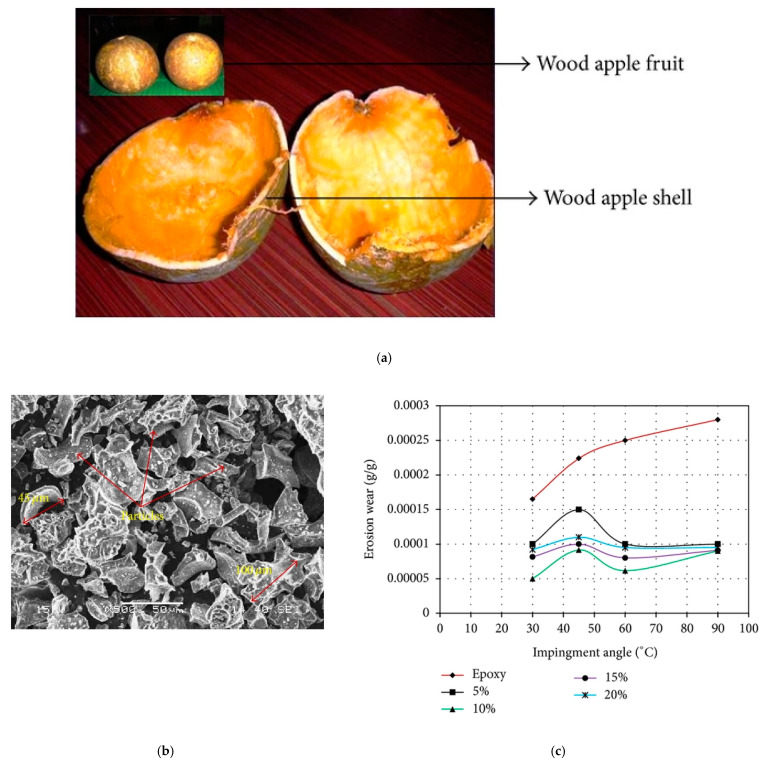
(**a**) Photograph of wood apple fruit and wood apple shell, (**b**) SEM images of the wood apple shell particles, (**c**) erosion wear behavior of the composites at different impingement angles [45]. Open access.

**Figure 6 polymers-14-00265-f006:**
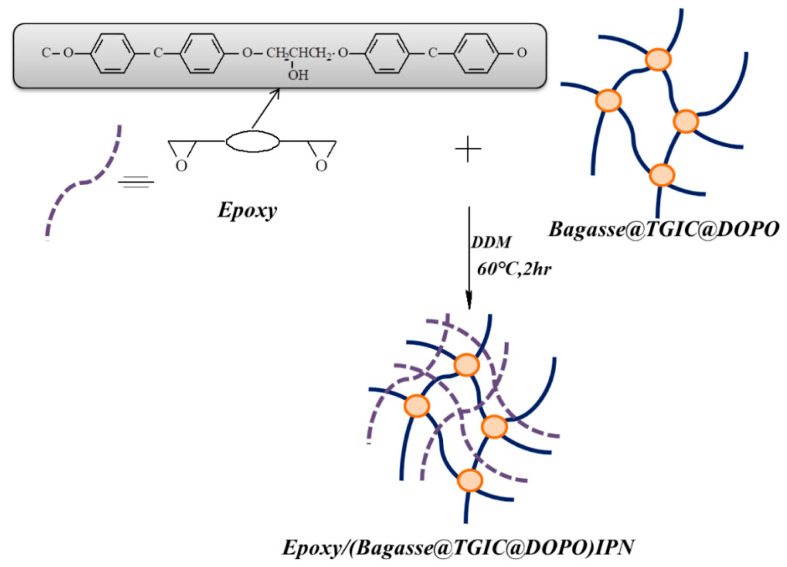
The reaction process of epoxy/bagasse @TGIC@DOPO IPN [50]. Open access.

**Figure 7 polymers-14-00265-f007:**
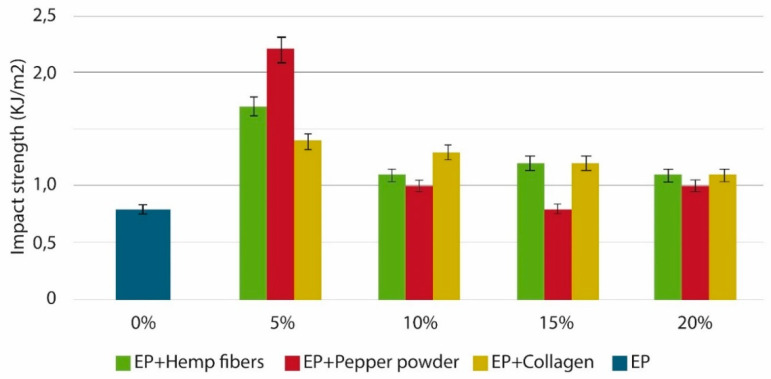
Impact strength of epoxy composites containing 5–20% natural filler (collagen, hemp fibers, and pepper powder [54]). Open access.

**Figure 8 polymers-14-00265-f008:**
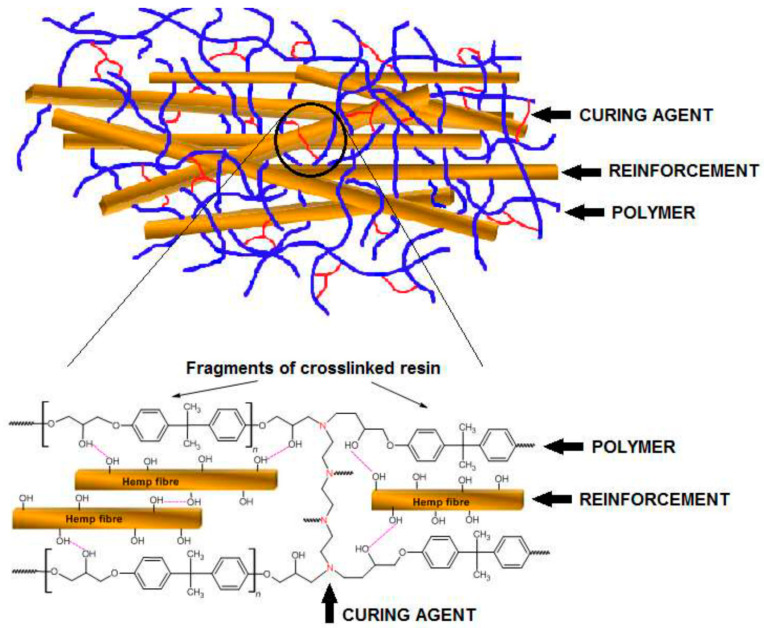
The schematic representation of intermolecular interaction between the hemp fiber, epoxy resin, and the curing agent (triethylenetetramine) [58]. (Open access).

**Table 1 polymers-14-00265-t001:** Fiber source, world production, cellulose content, tensile strength, Young’s modulus, and density of various natural fibers [17] (reproduced with thanks from Elsevier, License Number: 5206330256793).

Fiber Source	World Production (103 ton)	Cellulose (wt%)	Hemicellulose (wt%)	Lignin (wt%)	Waxes (wt%)	Tensile Strength (MPa)	Young’s Modulus (GPa)	Elongation at Break (%)	Density [g/cm^3^]
Bamboo	30,000	26–43	30	21–31		140–230	11–17		0.6–1.1
Bagasse	75,000	55.2	16.8	25.3	-	290	17		1.25
Coir	100	32–43	0.15–0.25	40–45		175	4–6	30	1.2
Pineapple									
Ramie	100	68.6–76.2	13–16	0.6–0.7	0.3	560	24.5	2.5	1.5
Abaca	70	56–63	20–25	7–9	3	400	12	3–10	1.5
Flax	830	71	18.6–20.6	2.2	1.5	345–1035	27.6	2.7–3.2	1.5
Jute	2300	61–71	14–20	12–13	0.5	393–773	26.5	1.5–1.8	1.3
Hemp	214	68	15	10	0.8	690	70	1.6	1.48
Sisal	378	65	12	9.9	2	511–635	9.4–22	2.0–2.5	1.5

## Data Availability

The data presented in this study are available on request from the corresponding author.

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
