# Peer review of "Natural Fillers as Potential Modifying Agents for Epoxy Composition: A Review"

_polymers, 2022, doi:10.3390/polym14020265_

Round 1

Reviewer 1 Report

The research work provides a good prospect on the effect of natural fillers on the properties of epoxy resin. The effects of different kinds of natural fillers on the mechanical properties of epoxy resin are also summarized in detail. It is suggested that the authors consider the following comments and add the necessary review summary for further improvement. 

  1. In the first paragraph of introduction, the authors mentioned that the epoxy resin has the advantages, such as thermal stability, water resistance, chemical corrosion resistance and dimensional stability. However, the above statement lacks of the preciseness and scientificity. As known, the epoxy group in resin is easy react with the water molecules to form hydrogen bonds, which further leads to a series of hydrolysis, plasticization of epoxy resin and interfacial debonding with the fibers. It is suggested that the authors refer to the following latest research work on the degradation of epoxy resin to rewrite the content. https://doi.org/10.1016/j.jmrt.2021.08.088

In addition, according to the above degradation summary of epoxy resin in the service environment, the authors should further propose the improvement ways of mechanical properties and long-term durability of epoxy resin through adding the natural fillers. Compared with mechanical properties, the long-term durability is the key evaluation index to determine whether epoxy resin can be used in engineering field.

  1. Page 2-3, the advantages of natural fiber are summarized in detail such as in figure 1 and table 1. However, the poor mechanical properties and water sensitivity of natural fibers should be further introduced. At the same time, the improvement of short-term and long-term properties of epoxy resin by adding natural fibers to epoxy resin should also be further mentioned.
  2. In the part of “Post agricultural waste powder material filled epoxy composites (seed and fruits)”, the authors summarized that seeds and fruits were filled into epoxy resin to improve the mechanical properties of epoxy resin. When the improved epoxy resin is exposed to moisture for a long time, what is the impact on the long-term performance of the epoxy resin when the seeds and fruits decay? How do you consider the above problems?
  3. For the part of “Grass fiber-based Epoxy composites”, the sentence of “Grass fibers have great potential to replace synthetic fibers” may be ambiguous. As we all know, synthetic fibers such as carbon fiber, glass fiber, aramid fiber and basalt fiber have very excellent mechanical properties, long-term durability and thermal resistance. Because of these excellent mechanical properties, it has a very favorable reinforcing effect on epoxy resin in mechanical properties, fatigue properties and durability. It is suggested to summarize the improvement effect of synthetic fiber on the performance of epoxy resin.
  4. On the synthetic fiber hybrid composites, the hybrid mechanism and its effect of the hybrid composites is unclear. Furthermore, the synergy effect between the fibers and the reinforcement effect to the epoxy resin should be analyzed. Please added the summary on the synthetic fiber hybrid composites, such as “carbon/glass fiber reinforced polymer composite”, “carbon/natural fiber reinforced polymer composite”

Polymer Testing, 2021, 104: 107384.

https://doi.org/10.1016/j.compstruct.2021.115060 

  1. The conclusion should be improved, including the key summary. 

Reviewer 2 Report

There are some weaknesses through the manuscript which need improvement. Therefore, the submitted manuscript cannot be accepted for publication in this form, but it has a chance of acceptance after a minor revision. My comments and suggestions are as follows:

1- Abstract gives information on the main feature of the performed study, but some details about the type of natural fillers must be added.

2- Authors must clarify necessity of the performed research. Objectives of this review study must be clearly mentioned in introduction.

3- The literature study must be enriched. In this respect, authors must read and refer to the following papers: (a) https://doi.org/10.4028/www.scientific.net/AMM.110-116.1361 (b) https://doi.org/10.1016/j.msec.2013.08.023 and other research works.

4- It would be nice, if authors could add some figures (real or schematic) to show concept and some conditions. Figures must be illustrated in a high quality (e.g., in Fig. 5 authors should type the legend)

5- The main reference of each formula must be cited. Moreover, each parameters in equations must be introduced. Please double check this issue.

6- The presented curves must be illustrated in a more scientific way. For example, standard deviation can be added.

7- Author must add a section and explain current challenges and limitation in this field with details and evidence.

8- In its language layer, the manuscript should be considered for English language editing. There are sentences which have to be rewritten.

9- The conclusion must be more than just a summary of the manuscript. List of references must be updated based on the proposed papers. Please provide all changes by red color in the revised version.

Round 2

Reviewer 1 Report

The authors have made necessary  modifications according to the relevant comments. I recommend accepting the paper.

Reviewer 2 Report

Dear Authors,

you have addressed the comments, and answered the questions. The revised version of your manuscript appears to be suitable for publication.

Best regards